# Atypical and Unpredictable Superficial Mycosis Presentations: A Narrative Review

**DOI:** 10.3390/jof10040295

**Published:** 2024-04-18

**Authors:** Zoubir Belmokhtar, Samira Djaroud, Derouicha Matmour, Yassine Merad

**Affiliations:** 1Department of Environmental Sciences, Faculty of Natural Sciences, Djilali Liabes University of Sidi-Bel-Abbes, Sidi Bel Abbes 22000, Algeria; zoubir_31@yahoo.fr; 2Laboratory of Plant and Microbial Valorization (LP2VM), University of Science and Technology of Oran, Mohamed Boudiaf (USTOMB), Oran 31000, Algeria; 3Department of Chemistry, Djilali Liabes University of Sidi-Bel-Abbes, Sidi Bel Abbes 22000, Algeria; 4Central Laboratory, Djilali Liabes University of Medicine of Sidi-Bel-Abbes, Sidi Bel Abbes 22000, Algeria

**Keywords:** tinea incognito, tinea corporis, hypopigmented skin, onychomycosis, dermatophytes, extensive dermatitis, mycosis fungoides, Majocchi

## Abstract

While typically exhibiting characteristic features, fungal infections can sometimes present in an unusual context, having improbable localization (eyelid, face, or joint); mimicking other skin diseases such as eczema, psoriasis, or mycosis fungoides; and appearing with unexpected color, shape, or distribution. The emergence of such a challenging clinical picture is attributed to the complex interplay of host characteristics (hygiene and aging population), environment (climate change), advances in medical procedures, and agent factors (fungal resistance and species emergence). We aim to provide a better understanding of unusual epidemiological contexts and atypical manifestations of fungal superficial diseases, knowing that there is no pre-established clinical guide for these conditions. Thus, a literature examination was performed to provide a comprehensive analysis on rare and atypical superficial mycosis as well as an update on certain fungal clinical manifestations and their significance. The research and standard data extraction were performed using PubMed, Medline, Scopus, and EMBASE databases, and a total of 222 articles were identified. This review covers published research findings for the past six months.

## 1. Introduction

Superficial mycosis is often difficult to identify and has various differential diagnoses especially at an early stage [1].

Misdiagnosis can easily occur when the lesions appear in unexpected ways, including at unusual ages, previously unaffected geographical areas, atypical body locations, or even unlikely contexts.

In addition, uncommon cases are related to an unpredictable course (short-term course), local or systemic treatments, and automedication [2], leading to dermatitis characterized by a high percentage of relapses [3].

Superficial mycosis can adopt an atypical presentation [3,4] especially among immunosuppressed individuals [5], and diagnosis is often delayed because of a lack of suspicion [6].

Dermatophytes often go undetected, resulting in continued spread of the fungus within individuals and to other susceptible individuals [7].

When atypical clinical features are present, tinea incognito is known to mimic several disorders, such as rosacea, cutaneous lupus erythematosus, and granuloma annulare [8]; it can present as lichenoid, eczema-like, psoriasis-like, and rosacea-like lesions, sometimes even bullous lesions [9].

Tinea incognito can also be misdiagnosed in a late stage. We must retain the fact that tinea is still a great mimicker of other dermatoses and not prescribe drugs without microscopic confirmation of dermatophytosis [9]. Moreover, diffuse tinea corporis can be misdiagnosed as psoriasis in a late stage [10].

Despite advances in laboratory techniques, special strains, and the molecular identification of fungi, history and clinical diagnosis remain paramount [1,9].

Due to the atypical presentations of superficial fungal infections in keratinization disorders, the nonspecific histopathological findings, and the frequent unsuccessful cultivation, achieving a correct diagnosis can be complex and requires re-evaluation of the condition and careful consideration [2,6,10,11,12].

To the best of our knowledge, no narrative review has been carried out evaluating the spectrum of atypical mycosis.

## 2. Factors Contributing in Atypical Presentations

The emergence of such a challenging clinical picture is attributed to a complex interplay of host, environment, and agent factors [5,13].

### 2.1. Atypical Epidemiological Context

The change in the prevalence of fungi causing atypical disease can be linked to geographic location, exposure to a large amount of spores due to warm and humid climate, and hosts characteristics (culture, habits, and migration). Moreover, spores may remain viable in suitable environments for up to 12–20 months, and some spores were also reported to persist for at least a year in salt water. Certain types of spores (e.g., microconidia) might be dispersed via airborne means [14]. High-exposure locations such as swimming pools, nails salons, and wrestling mats can also induce dermatophytosis [15].

#### 2.1.1. Environmental Factors

Fungi are found universally. However, the ecological landscape plays a crucial role in shaping the fungal threat. Their abundance and virulence potential fluctuate across different niches and geographic areas, with dermatophytes serving as a prime example, which are more common in tropical regions [16,17]. Effectively, hot, humid climates favor fungal growth, leading to a rise in these infections in tropical and developing countries [18]. However, an experimental study found that neither freezing dermatophytes at −20 °C for 24 h or one week nor exposing them directly to heat at 60 °C for 10, 30, or 90 min was effective in killing dermatophyte conidia [19].

Some fungi are spread globally while other rare pathogens are located in specific areas. In Bangladesh and Inda, superficial mycoses are masked by overlapping symptoms from diseases like tuberculosis, chronic inflammation, and malignancies. This underdiagnosis and underreporting paint an incomplete picture of the true fungal burden. Studies shed light on how some atypical fungal infections, like “connubial dermatophytosis” in spouses and extensive “tinea corporis” in infants, thrived within the close quarters of families [5,20,21].

It is extremely uncommon for travelers returning from endemic regions to have tinea imbricata. However, a case of tinea imbricata in an Italian child was reported after a trip to Solomon Island. This form of chronic tinea corporis caused by *Trichophyton concentricum* was described as a squamous, widespread, concentric lesion [22].

#### 2.1.2. Fungal Factors

Superficial mycoses are caused by yeasts, dermatophytes, and, less commonly, by molds. The number of fungal cases is constantly increasing and many new fungal pathogens are being identified [14]. Fungi can be classified as primary or opportunistic pathogens based on their virulence [14].

The geographical distribution of dermatophytosis varies by species. Four fungal species—*Trichophyton rubrum*, *Trichophyton mentagrophytes*, *Trichophyton tonsurans*, and *Microsporum canis*—have become increasingly common worldwide, establishing themselves as the leading cause of infections. In contrast, other dermatophytes, like *Trichophyton verrucosum*, *Trichophyton violaceum*, and *Microsporum ferrugineum*, remain more prevalent in specific regions of Europe, Asia, and Africa. Furthermore, *Trichophyton tonsurans*, is more commonly found in the United Kingdom and North/South America [23]. On the other hand, studies have shown that *Trichophyton indotineae*, which originated in the Indian subcontinent, has spread to numerous countries worldwide [24].

*Trichophyton rubrum* is the fungus most frequently involved in atypical forms [25,26,27] associated with a strong tendency for dissemination and that are usually refractory to topical and systemic therapy [26]. Moreover, research suggests that *Trichophyton rubrum* is the primary fungus responsible for chronic dermatophytosis infections [18].

In addition, uncommon cases of *Trichophyton tonsurans* infection have been previously reported [28], causing atypical tinea corporis in Algeria where this species is not commonly recovered. Furthermore, this dermatophyte can occasionally be responsible for tinea pseudoimbricata, manifesting as multiple concentric annular erythemas [29,30], especially when long-term topical steroids misuse occurs [5,30,31,32].

*Trichophyton indotineae* is another lesser-known species that has recently been reported as a difficult-to-treat dermatophytosis [33].

Interestingly, *Nannizzia nana*, reported previously as *Microsporum nanum*, has no specific clinical picture [13] and induces onychomycosis of toenails as well as skin lesions. Moreover, improved diagnostic tools have led to an increase in identified *Nannizzia nana* infections, suggesting that misdiagnosis was previously masking the true prevalence rather than a low infectivity of this agent [2].

Traditionally, diagnoses focused on well-known fungal culprits. But, with improved tools (PCR and MALDI-TOF MS), we are now recognizing a wider spectrum of less common fungal species capable of causing superficial mycoses.

Usually, zoophilic species are characterized by more pronounced inflammation than anthropophilic species, with the formation of vesicles [34,35]. Some fungi can act opportunistically, causing severe and unusual infections [3]. *Candida parapsilosis* association with environmentally acquired skin ulcers can resemble cutaneous sporotrichosis [36]. While *Malassezia* species typically reside harmlessly on the skin, they can trigger inflammatory responses leading to symptomatic skin conditions (folliculitis, dandruff, and eczema) in both humans and animals [37].

Rare fungi are also responsible for unexpected skin conditions. *Prototheca* sp. are ubiquitous, commonly isolated from grass, soil, and water; this species is responsible for cutaneous infections [38]. *Emmonsia crescens* formerly known as *Chrysosporium parvum* var. crescens is a saprophytic soil fungus transmitted via inhalation and inducing rare pulmonary diseases and exceptional cutaneous localization [38].

Finally, immune status can specifically play a role in the virulence of some fungal species. A previous study discovered that the rate of dissemination among patients with blastomycoses was consistent across the immunologic spectrum, which is in deep contrast to other endemic fungi such as histoplasmosis. This suggests that pathogen-related factors have a greater influence on dissemination for blastomycosis than immune defense [39].

#### 2.1.3. Host Factors

Numerous host factors increase susceptibility to fungal infection, such as comorbidities (obesity, diabetes mellitus, immunosuppressive disorders, and poor circulation) [14], socioeconomic background, lifestyle changing, poor hygiene (sweaty clothing and bedding), poverty, and occupational status [5,13,14,40]. Even diet has been suggested as a factor that contributes to adult kerion associated with onychomycosis in vegetarian individuals [41].

The origin of the patients can influence the clinical picture; a surprising highly significant occurrence of tinea capitis due to anthropophilic species was revealed among Spanish school children from an African immigrant population [42]. Population migration affects the spread and diversity of fungi [43,44]. Emergent fungi, like *Trichophyton tonsurans*, are increasing in Europe, particularly due to the growing immigrant population [42,45], and beyond, illustrating how human–animal interactions, including pets, farm animals, and wild animals can reshape the fungal landscape [13].

Some superficial fungal infections are uncommon in certain age groups. The clinical pictures of superficial fungal infections in children can be diverse, non-distinctive, and somewhat confusing [46]. Tinea capitis is the most frequent dermatomycosis in childhood; however, it is rare in newborns [43,45,47]. There have been unusual cases reported where the newborn had a thick, scaly adherent mass on their scalp without any suppurations or alopecia.

Tinea capitis is unusual in adults; nevertheless, it seems that this mycosis is not so rare, especially in immunocompetent women who spent their childhood in Africa [45]. Similarly, Tinea unguium and tinea pedis have been rarely reported in prepubescent children [48,49,50].

Onychomycosis is more common among adults and elderly with an increasing prevalence with age due to reduced nail growth accompanied by an increase in nail plate thickness. Thumb sucking is a notable behavior in newborns and infants, inducing maceration of the fingers and increasing the risk of oral flora infection to the nail folds and the hyponychium [51].

##### Trauma

There are several types of trauma, including accidental trauma, self-induced trauma, non-conventional treatment, and modern medical procedures.

Dermatophytosis could probably be initiated by physical trauma, serving as gateway for *Trichophyton rubrum* [8]. Reportedly, a woman who shaved her legs developed a skin condition that was misdiagnosed as mycosis fungoides but was in fact a form of dermatophytosis [1]. Furthermore, an adult healthy female presented several axillary and perineal ulcers following incision and drainage of slowly growing nodular lesions over a one-year duration. She admitted to shaving her axillae and pubic region with a safety razor several times. Both culture and histology revealed an important presence of *Aspergillus* [52].

Traditionally, thorn injuries are produced by *Sporothrix schenckii*, yet *Candida parapsilosis* has also been implicated following rose thorn injury in an immunocompetent patient, and the wound healed following itraconazole medication [36]

Ear Trauma: Some traumas are induced by the patient’s habits, such as ear self-cleaning leading to otomycosis. Moreover, self-inducted injuries are already described in schizophrenia, and *Aspergillus flavus* otomycosis has been linked to ear self-mutilation [53,54] (Figure 1).

Cutaneous injury, agricultural trauma, traffic injuries, and orthopedic trauma have all been linked to primary cutaneous aspergillosis [55,56,57]; usually, the symptoms appear within one month of injury [57]. Moreover, burn wounds can be infected by Aspergillus species [55,58]; Panke et al. described characteristic “Aspergillus fruiting bodies” on the skin of a burned patient [59].

##### Underlying Diseases

Atypical mycosis is conditioned by host susceptibility to fungal infection. Associated cutaneous, systemic, or iatrogenic disorders along with immune dysfunction should be taken in account in individuals with recalcitrant dermatophyte infections [11,60,61,62]. Atopic persons and those infected with zoophilic fungi tend to have more inflammation [35]. Atypical morphology of superficial fungal infections is a characteristic feature of keratinization disorders [12].

Delayed keratin scaling in atopic disease may facilitate the persistence of fungal infection [63], mimicking the appearance of eczematous lesions, and potentially leading to misdiagnosis [63,64]; moreover, ichthyosis create a favorable environment for fungal growth and proliferation, since abundant keratinized cells provide a rich source of nutrients for dermatophytes [63,65].

Additionally, the more alkaline environment found on the skin of atopic eczema patients enhances the release of allergens from *Malassezia sympodialis*. Nanovesicles from *Malassezia sympodialis* and host exosomes induce cytokine responses—novel mechanisms for host–microbe interactions in atopic eczema [66].

When patients originally present with generalized typical skin condition like psoriasis or localized nevus the dermatologist can assume that all lesions are the same, while association with fungal infection is still possible [67,68]. Widespread tinea corporis with psoriasis vulgaris can lead to diagnostic confusion [67].

Reportedly, rheumatoid arthritis lesions on prednisone led to a leg rash due to *Purpureocillium lilacinum* [69].

Atypical, invasive, and disseminated clinical presentations are more likely seen in immunocompromised patients [70,71], including those with HIV [70,72,73]. A disseminated *Fusarium oxysporum* infection with skin localization on both calves was diagnosed in a 32-year-old female with relapsing B-acute leukemia during induction chemotherapy [74]. Atypical inflammatory bullous lesions due to *Trichophyton mentagrophytes* have been described in HIV individuals [75]; additionally, dermatophytosis mimicking Kaposi’s sarcoma has been documented in HIV patients [76,77]. Deficiency in caspase recruitment domain-containing protein 9 (CARD9) has also been shown to be associated with more severe presentations of tinea infections [78]. Primary and metastatic malignancies may occasionally mimic or coexist with cutaneous fungal infections [79].

Consequently, recognizing uncommon fungal infections as well as fungal cases presenting clinical patterns with other dermatoses is critical for immunocompromised patients to ensure prompt and adequate management [69,70].

##### Iatrogenic Factors

There are some treatment alternatives that are lengthy, aggressive, or have significant negative effects [1]. It is highly probable that cosmetic products alter the quantity and/or quality of local sebum secretion, creating an environment that promotes infection by *Microsporum canis* [80].

The self-medication habit of indiscriminately using steroids, antibiotics, and zinc during the COVID-19 pandemic may have contributed to the dysbiosis of gut microbiota and increased the risk of mycosis—specifically Mucorales with their unpredictable and severe cutaneous manifestations [81].

Corticosteroid use combined with antifungal medication may increase the risk of *Trichophyton indotineae* infection. In vitro studies have shown that point mutations in the squalene epoxidase gene can lead to terbinafine resistance, resulting in difficult-to-treat tinea corporis, tinea cruris, and tinea faciei [24].

##### Misuse of Treatment

The majority of misdiagnosed dermatophytosis cases are mistaken for bacterial infections or infected eczema and are, consequently, treated with various antibiotics and corticosteroids without effect [82].

Previous treatment with topical corticosteroids might justify some patients being erroneously considered as having Lupus erythemateous [83]. A report on a patient having pityriasis versicolor with clinical and mycological evidence of palmar lesions and fingernail onychomycosis was strongly related to mishandling of *Malassezia furfur* infection [84].

Misuse of treatments can contribute to fungal resistance and potentially mask the true nature of a typical mycosis [1]. Several drugs may be involved, as follows:

##### Indigenous Medicines

The use of indigenous medicines has been cited as factor favoring atypical forms of dermatophytosis in India. The application of uncontrolled topical preparations can distort the initial lesions [5].

##### Medical Drugs

Corticosteroids

Typical ringworm symptoms are masked by previous topical and/or systemic corticosteroids, alongside the concomitant application of emollients [2,85,86]. Unusual manifestations of tinea corporis are known as “Tinea incognito”, which is commonly used for steroid-modified cases of dermatophytosis, although later reports also show the use of tacrolimus and pimecrolimus for dermatophytosis [5].

Non-diabetic COVID-19 patients, particularly those who were given high doses of steroids for a long time, developed mucormycosis [81].

Deferoxamine

Individuals receiving deferoxamine medication are especially vulnerable and they can develop cutaneous manifestations of Mucorales [81]. Deferoxamine reportedly aggravates skin and sino-orbital mucormycosis [87,88].

Antithymocyte globulin

Antithymocyte globulin was associated with skin aspergillosis among patient treated for agranulocytosis [89].

Immunosuppressive drugs

IL17-inhibitor, tocilizumab, pimecrolimus, and tacrolimus have been linked to tinea incognito [5,10,90].

Chemotherapy

Docetaxel chemotherapy administered to breast cancer patients induced *Candida guillermondii* onycholyse associated with exudate and oedema [91] (Figure 2).

##### Medical Procedures

Bandage/gauze

Cutaneous aspergillosis is typically found when gauze or bandages come into contact with the skin [92].

Ventilators

Patients on oxygen/ventilator support may incur severe cases of nosocomial mucormycosis [81]. Invasive tracheobronchitis caused by Mucorales or other molds has been reported in manually ventilated patients in ICUs, with a 93.5% overall mortality rate. High-flow nasal cannula delivers a high, humidified, and heated flow; this can occasionally cause nasal mucosal damage. On the other hand, paranasal sinuses provide a suitable environment for rhino–orbital–cerebral mucormycosis, leading to necrosis and superficial eschars. Furthermore, while only distilled water should be used for hydrating oxygen, sometimes tap water, or any other available water, is being used either due to ignorance or negligence [81,93,94].

Catheters

A cutaneous lesion behind a venous catheter’s transparent dressing was reported by Smith and Wallace [95].

Skin necrosis around the removed Hickman line site was followed by other similar lesions around the arterial and venous cannulation sites [96].

Injection

Unexpectedly, a 37-year-old woman developed granulomatous lesions on their cheek caused by *Trichophyton rubrum* after facial injection with hyaluronic acid [97].

Organ Transplantation

Patients undergoing liver or kidney transplants may develop primary cutaneous aspergillosis immediately in the surgical site [98] or acquire tinea incognito [81].

## 3. Atypical Context of Infection

A surprising nosocomial tinea corporis has been described in a HIV patient hospitalized for Cryptococcus meningitis, after an unexpected report of cat transmission [99]. (Figure 3).

Unexpected otomycosis among earmold wearers is possible since the device offers an occlusive environment, thereby facilitating fungi growth. On the other hand, self-mutilation in schizophrenia could lead to surprising filamentous otomycosis [53,100].

Tattoos can have several inflammatory complications including allergic contact dermatitis, lichenoid, photo-induced and granulomatous reactions, pseudolymphoma, and skin cancers or infectious complications such as bacterial infections (leprosy, syphilis, pyoderma, mycobactriosis, and cutaneous tuberculosis), viral infections (molluscum contagiosum, hepatitis B and C, and herpes simplex), and fungal infections (dermatophytosis and sporotrichosis) [101]. Injury to the skin caused by a tattoo needle, or contaminated ink, or the use of non-sterile instrument, and/or contact with a contaminated animal or human source are the principal etiologies of this particular form of dermatophytosis. Tinea may appear either early during the first two months following tattoos, or later in a healed tattoo [102]. The main dermatophytes isolated from tattoos included *Trichophyton rubrum*, *Microsporum canis*, *Trichophyton tonsurans*, *Epidermophyton floccosum*, and *Microsporum gypseum* [102].

Penile shaft dermatophytosis is a rare form described among Indian male wearers of ‘lengoty’; the semi-occlusive dress can unexpectedly facilitate the fungal growth [103].

## 4. Atypical Clinical Features

Emerging atypical and unusual presentations are widely described in developing countries [5].

Atypical mycosis infections should be treated promptly and properly, otherwise they can become chronic and require oral antifungal medication [104].

In tinea incognito, the lesion on the face can be eczema-like, rosacea-like, or discoid lupus, while the trunk and the extremity lesions can be impetigo-like [105,106]; examples of atypical dermatophytosis are summarized in Table 1.

Tinea incognito is a rare form of dermatophytosis compared to the more common presentations [1]. It often presents with multiple lesions at various sites, exhibiting varying degrees of inflammation [123]. For instance, *Trichophyton indotineae* triggers itchy and inflamed skin infections that can spread across the groin, buttocks, torso, and face [24].

Atypical cases of *Microsporum canis* were described as nummular eczema, erythema multiforme, granuloma annulare, granuloma faciale, and lymphocytic infiltration of the skin [124]; pityriasis rosea and seborrheic dermatitis [124,125,126], and lupus erythematosus [83,117].

In a multi-center study including 283 tinea incognito patients in Korea, the following diagnoses had been incorrectly applied to tinea incognito patients: nonspecific eczema, atopic dermatitis, contact dermatitis, seborrheic dermatitis, diaper dermatitis, intertrigo, nummular dermatitis, stasis dermatitis, psoriasis, lupus erythematosus, urticaria, and lichen simplex chronicus [90]. Another study presenting 200 cases of tinea incognito revealed that the most common resembling dermatoses were psoriasis, rosacea, impetigo, discoid dermatitis, lupus erythematosus, polymorphic light eruption, seborrheic dermatitis, lichen planus, and erythema migrans [114].

Uncommon retinochoroiditis secondary to fungal infection of the skin caused by Tinea corporis has also been reported [127]; the patient had extensive reddish, erythematous, plaque-like skin lesions over the abdomen and back.

-Psoriasis-like

Both large plaques with silvery scales and less infiltrated lesions with fewer scales have been seen in cases resembling psoriasis [67,68]; the lesions were commonly located on lower extremities, trunk, or palms [68,107,108].

Tinea corporis caused by *Trichophyton rubrum* was reported to mimic a flare-up of psoriasis under treatment with IL17-inhibitor Ixekizumab [10]. Moreover, pustular psoriasis-like tinea incognito due to *Trichophyton rubrum* was described [86]. Furthermore, a case of *Microsporum canis* in an elderly woman was initially diagnosed as psoriasis and seborrheic dermatitis [128].

The patient’s past medical history of psoriasis vulgaris would cause misdiagnosis and delay in the treatment of dermatophytosis [129]. A case of concomitant psoriasis and tinea capitis due to *Microsporum canis* occurred in a child; the clinical picture was more severe than that depicted in Figure 4.

-Atopic-like

*Trichophyton rubrum* has been incriminated as a possible trigger in flare-ups of atopic dermatitis [130].

Patient’s atopic dermatitis infection can also be exacerbated by chronic dermatophytosis [131].

-Eczema-like

Eczema and tinea, despite being distinct skin conditions, share the common feature of causing dry, inflamed skin. These presentations are seen in immunosuppressed as well as immunocompetent individuals [123]. In Italy, 82% of the tinea incognito cases observed were eczema-like presentations [90]. Dutta et al. also found that eczema-like lesions were the most common [112].

Atzori et al. observed eczema-like tinea as the most common atypical manifestation of tinea [132].

A case of a fungal skin infection initially mistaken for eczema was later diagnosed as biopsy-confirmed CTCL and successfully treated with antifungal medication [1].

-Seborrheic dermatitis-like [124,125,126]

Seborrheic dermatitis-like tinea capitis that was misdiagnosed in an elderly Chinese man, with samples taken from the scalp and nails, revealed the presence of *Trichophyton rubrum* with an association of tinea capitis and tinea unguium [116].

An elderly woman presented seborrhoeic dermatitis which was diagnosed later as a dermatophytosis due to *Microsporum canis* [128].

-Acne-like

Malassezia fungi can also cause folliculitis, primarily affecting sebum-rich areas like the face, chest, and back. This condition mimics acne but does not produce comedones [133].

Although sterile folliculite-like lesions are found on torso or back of patients with Behcet’s disease, lesions acne-like due to *Trichophyton rubrum* were described on the back of a patient receiving corticosteroids therapy [118] (Figure 5).

-Lupus Erythematosus-like

The misdiagnosis of tinea mimicking lupus erythematosus has been reported in the literature; the great majority of cases are tinea faciale with some exceptions involving the trunk. Some cases coexist with true lupus erythematosus [83]. Tinea is confirmed by the presence of hyphae in systemic LE-like eruptions, the dermatophytes responsible for such infections are varied and *include Trichophyton rubrum*, *Trichophyton mentagrophytes*, *Trichophyton tonsurans*, *Trichophyton verrucosum*, and *Microsporum canis* [83,117,134,135].

-Sweet’s syndrome-like

A 53-year-old female with a 2-month history of a painful itchy rash on her face had previously presented with multiple edematous pseudo-vesicular papules that appeared suddenly on the face and shoulders, followed by a febrile upper respiratory tract infection simulating sweet’s syndrome [16].

-Ichtyosis-like

Despite the prevalence of both dermatophytosis and ichthyosis in clinical practice, their concomitant occurrence is seldom documented [63].

-Keloid-like

A 53-year-old man presented with a rapidly growing plaque on their right forearm. He worked as a seasonal olive harvester and may have unknowingly been scratched by an olive branch. The lesion had a keloid-like or brainy appearance. A mycological study identified *Trichophyton rubrum* [136].

-Leproy-like

Hypopigmented scaly patches on the body, simulating leprosy [20].

-Rosea-like

Rosacea-like tinea incognito [32].

### 4.1. Atypical Localization

Superficial fungal infections of the hair, skin, and nails are a major cause of morbidity in the world [137].

Improbable localizations of mycoses can lead to misdiagnosis. Depending on different clinical manifestations, dermatophytosis is classified using the anatomical region where it typically occurs, such as tinea unguium, tinea pedis, tinea cruris, tinea corporis, and tinea capitis [11].

It is important to consider a dermatophyte infection when dealing with dermatosis that may be in an atypical area given the age of the patient or have an appearance that may be uncommon for a fungal infection [11], examples of atypical localizations of superficial mycosis are depicted in Table 2.

When the host is immunocompromised, the infecting organism may be an opportunistic fungal organism or a dermatophyte that would generally not be expected to be recovered from that site [11].

*Trichophyton violaceum* was responsible for concomitant tinea of the scalp and eyebrows in a 62-year-old diabetic woman [11].

*Trichophyton verrucosum* often causes inflammatory, deeply infiltrating lesions located on the head, face, hands, and forearms, but aphlegmasic, superficially spreading lesions affecting other parts of the body have also been reported [144].

Atypical dermatophytosis progression without a clearing center encourages inflammatory, crusting forms; thickness of the corny layer; and uncommon anatomical localizations including sebaceous glands, skin folds, and vellus hair follicle [5].

-Facial involvement

Children including infants with facial involvement and adults with tinea capitis have become a frequent clinical presentation in India [5]. In children, Malassezia-related conditions and atypical facial involvement, particularly on the temples, can be observed, and fine, white, flaky scales may also be present [133].

An erythematous nodule on the lip due to *Trichophyton rubrum* was reportedly following a trauma in the same area [143].

Ocular involvement due to dermatophytes can present as eyelid infestation [11].

-Flexural involvement

Infection located at joint can be attributed to psoriasis; there is little documentation regarding the unusual pityriasis versicolor localizations of the body like the scalp, face, arms, palms, legs, soles, flexural areas, areolae, and penile involvement [138].

The term tinea axillaris has been used only a few times in the literature; it is considered as a variant of intertriginous tinea—a unique non-occupational *Trichophyton verrucosum* tinea axillaris has been described [144].

Unusual tinea versicolor of the axilla can appear as well-demarcated erythematous lesion with minimal scale [139].

-Ungual involvement

Toenails are 25 times more likely to be infected than fingernails as the causative molds are ubiquitous fungi seen in soil, water, and decaying vegetation [145]; fingernail localization is more confusing.

-Genital involvement

The base of the penis is most often affected by dermatophytes, followed by the shaft and the prepuce [123].

Atypical pityriasis versicolor forms include penile involvement [138] and genitalia [138,146,147].

Tinea infection has been uncommonly described on the penile shaft [103].

-No evident localization

Allergic dermatophytid is a distant reaction to dermatophytes, it is an allergic rash that will disappear once the original infection has been treated.

### 4.2. Atypical Progression and Extension of the Lesions

Most of the fungal cutaneous presentations are typically benign and easily treated [78].

Disseminated forms are not linked to age; therefore, an extensive form in a neonatal patient was attributed to dermatophyte [148].

Reportedly, deep and extensive forms can indicate a more serious underlying immunodeficiency [11,70,78,149], including those with HIV [70,72,73]. However, a case of onychomycosis involving all 10 fingers of an immunocompetent male with no comorbid conditions was caused by filamentous *Aspergillus niger* [145].

The failure of patients and clinicians to recognize a fungal infection early may lead to a more extensive, severe, and difficult-to-treat disease [2,150]. Indeed, the delay in diagnosis and application of topical corticosteroids also contributed to lesion generalization [128].

The clinical description of tinea incognito is highly variable. Compared with the lesions of tinea corporis, the lesions seen in tinea incognito are generally less erythematous and scaly, less well-defined, and generally more extensive [105,106].

It is hypothesized that extensive forms are related to some fungal species. For an instance *Microsporum canis* transmitted from animals has stronger pathogenicity which makes the skin lesion more generalized [128]. Additionally, widespread tinea corporis is resistant to oral antifungals and may not respond well to them, particularly if immunosuppression is present [60].

An elderly woman with well-controlled diabetes mellitus presented with a six-month history of erythema with yellow crusts on her scalp and extensive erythematous patches with scales on the body skin caused by *Microsporum canis*. The patient’s condition was initially diagnosed as psoriasis and seborrheic dermatitis [128].

Fungal rashes are usually harmless and respond well to topical antifungal treatments, more extensive or persistent manifestations could signal an underlying immune deficiency that requires further investigation [78].

Widespread tinea corporis are recalcitrant and could be poorly responsive to oral antifungals, especially if immunosuppression is associated [60], and they present as slightly scaly eruption.

Majocchi’s granuloma, also known as nodular granulomatous perifolliculitis, results from fungal penetration throughout the hair follicle to the dermal or subcutaneous tissue, causing a suppurative folliculitis [105].

Tinea incognito refers to cutaneous fungal infections that have lost their typical morphological features due to the use of calcineurin inhibitors or corticosteroids [105,106]; the delay in diagnosis and application of topical corticosteroids also contribute to lesion generalization [128]. The clinical presentation of tinea incognito is highly variable. Compared with the lesions of tinea corporis, the lesions seen in tinea corporis are generally less erythematous and scaly, less well-defined, and generally more extensive. The lesion on the face can be eczema-like, rosacea-like, or discoid lupus, while the trunk and the extremities lesions can be impetigo-like [105,106].

The fungal species can also contribute to the spread of the lesions; some extensive forms of pityriasis were attributed to *Malassezia globosa* whose pathogenicity has been attributed to high lipophilic activity due to the levels of lipases and esterases [114,151]. Moreover, *Microsporum canis* that is transmitted from animals has stronger pathogenicity, which makes the skin lesion more generalized [128]; for instance, tinea Capitis by *Microsporum canis* in an elderly diabetic female with gradually extensive infection has been described, including dermatophyte infection of the scalp, whole trunk, groin, and all extremities with severe pruritus and malaise. The patient’s condition was initially diagnosed as psoriasis and seborrhoeic dermatitis and treated with econazole nitrate and triamcinolone acetonie cream without improvement [128]. Furthermore, a reported extensive tinea corporis due to *Trichophyton schoenleinii* in an 80-year-old woman on her forearms, thighs, legs, buttocks, and trunk was mimicking parapsoriasis, without scalp involvement [152].

Widespread tinea corporis and unguium has been shown to affect both axillary regions. *Trichophyton verrucosum* was isolated as the causative agent [144].

Immunocompromised patients (diabetes mellitus, chronic liver and kidney disorders, and transplant recipients) are also more likely to develop deep dermatophytosis, a rare disorder characterized by the invasion of dermatophytes into the dermis and subcutaneous tissue, rather than just keratinized skin [71].

## 5. Atypical Clinical Forms

### 5.1. Atypical Onychomycosis

Reports of onychomycosis in prepubescent children are rare [48,49], this rarity has been attributed to the differences in nail plate structure, less exposure to trauma, and faster linear nail growth. Oral colonization by pathogenic yeasts and finger suckling could be potential risk factors for unexpected neonatal onychomycosis. Uncommon cases were attributed to *Candida albicans*, *Candida parapsilosis*, *Candida tropicalis*, *Trichophyton rubrum*, and *Fusarium oxysporum* [51,153,154]

Toenail changes are frequently observed among the subset of patients with vascular disease and chronic leg ulcers; nail modifications may also be the result of microangiopathy and subsequent chronic ischemia. Onychomycosis can also be confused with dystrophic toenails from repeated low-level trauma [155].

Uncommon onychomycosis clinical features such as onycholysis, oedema, and exudate have been linked to chemotherapy in breast cancers patients [91]. Melanonychia of fungal origin with brown or black pigmentation of the nail unit is a relatively rare clinical sign and can mimic subungual melanoma [156,157,158,159,160].

Both Aspergillus and other molds are an emerging cause of onychomycosis, mainly affecting the toenails of diabetic individuals [157,160,161]. The most common causes of melanonychia are dematiaceous, melanin-producing molds—*Scytalidium dimidiatum* and *Alternaria* sp. [156,157]. Moreover, subungual onychomycosis due to *Aspergillus niger* can mimic a glomus tumor [162].

Furthermore, blackening of nails with pits, could be a result of the coexistence of psoriasis and onychomycosis [163].

Additional rare clinical deformities include the so-called “pincer nail deformity”, known as “incurvated nail”, which is a transverse overcurvature of the nail [164]. Although the underlying pathogenesis is not clearly understood, this condition can hide an onychomycosis [164]. Moreover, onychogryphosis, known as ram’s horn nail, is a rare disorder observed in elderly people. It is characterized by dark thickening of the nail plate; *Aspergillus niger* has been implicated as a possible causal agent [160] (Figure 6).

Even the well-known yeast *Candida albicans*, rather than any other *Candida* species, often clinically presents as a misleading paronychia, or onycholysis [153]; some of the unusual forms of onychomycosis are depicted in Table 3.

### 5.2. Atypical Hair Mycosis

Tinea capitis is a common condition among children; nevertheless, it has been unexpectedly described in newborn [46,47,168] and adults [41]. The increasing occurrence of adult tinea capitis is probably linked to population aging and immune system modifications induced by diseases such as malignancy, diabetes mellitus, and long-term use of glucocorticoids and immunosuppressants [116].

Compared with children, adult tinea capitis is more often associated with another superficial mycosis caused by anthropophilic fungi such as *Trichophytotn rubrum*, *Trichophyton violaceum*, and *Trichophyton tonsurans* [116].

*Trichophyton rubrum* Seborrheic-like lesions of the scalp were described in a 77-year-old Chinese male patient [116], the patient experienced a pruritic scaly scalp, with increased hair loss over 6 months. In addition, the patient was suffering from associated tinea pedis and onychomycosis [116]. Furthermore, tinea capitis can be confused with alopecia areata [155].

#### Piedra

These infections are prevalent in tropical regions and appear as soft, attached nodules on scalp hair, but they can also affect the axilla, pubis, and beard areas. These nodules can easily be mistaken for nits, hair casts, or other hair conditions [133,169]. The differential diagnosis of piedra includes chronic intertrigo, hair casts, monilethrix, nits, pediculosis capitis, pediculosis pubis, tinea capitis, trichomycosis axillaris, trichoptilosis, and trichorrhexis nodosa.

### 5.3. Atypical Skin Mycosis

#### 5.3.1. Atypical Tinea Faciei

Tinea faciei is uncommon and often misdiagnosed at first; it is frequently aggravated by sun exposure. It may also present as a kerion (fungal abscess). Therefore, it is a disease nearly exclusively found in adult males. The presence of this kind of lesion on the same areas in women and children is classified as tinea faciei [170].

Tinea barbae is an uncommon superficial dermatophyte infection of the beard and moustache areas; it can mimic many other skin disorders—iododerma, contact dermatitis, perioral dermatitis, and actinomycosis [170].

Like tinea capitis, tinea barbae should be treated with oral therapy; systemic anti-fungal medications are able to penetrate the infected hair shaft whereas topical therapies cannot [170].

Tinea faciei due to *Trichophyton mentagrophytes* has been reported in an acromegaly patient under corticosteroids treatment (Figure 7) [171].

Tinea auricularis was used to design an ear infection treatment for the species *Microsporum canis* and *Trichophyton rubrum* [86,123]. Tinea labialis is also rarely used.

#### 5.3.2. Atypical Breast Localization (Tinea Mammae)

Tinea mammae (fungal infection of the breast) has been rarely reported [172]. These are unusual locations to find a tinea infection, particularly in someone with a healthy immune system [11].

Candidiasis in a breastfeeding mother is also a rare form of fungal nipple disease [173] (Figure 8).

Similarly, unilateral tinea mammae simulating atopic eczema with gradual increasing size due to *Trichophyton rubrum* has been described in an elderly man [172]. On the other hand, bilateral and symmetrical dermatophytosis can be found in adults [174]; one of the possible explanation of bilateral Tinea mammae might be the use of a contaminated bra [174].

A 28-year-old female presented with a prominent, reddish lesion in her left breast that had grown during the preceding three months and became nodular with yellowish discharge. Surprisingly, the mycological examination was compatible with *Trichophyton violaceum* [175].

Reportedly, bilateral areolar and periareolar pityriasis versicolor were documented [176,177].

#### 5.3.3. Atypical Tinea Manuum and Tinea Pedis

Tinea pedis and tinea manuum were previously considered uncommon fungal infections in children before puberty [50]. Emerging evidence from dermatological studies indicates that fungal infections of the feet and hands are more prevalent in children than previously thought [45,48,50]. Moreover, asymptomatic tinea pedis in children has been documented [178].

Some occupations involve handling fruits and vegetables and contact with these products without wearing gloves could be a predisposing factor for mycosis [145,158] (Figure 9).

Tinea manuum is less common than tinea pedis [34,35]. Typically, the infection affects one hand and both feet, or both hands and one foot [35].

Both fungal infections and hand eczema can manifest with similar symptoms, including red, itchy rashes. Hand eczema is commonly bilateral and symmetrical, whereas tinea manuum typically affects just one hand. Moreover, tinea manuum can present with nail involvement, while nails are not involved in hand eczema. Tinea pedis, also known as athlete’s foot, manifests as itchy, red, and inflamed areas on the feet. These lesions can appear on the soles (vesicular type), sides (moccasin type), or between the toes (interdigital type) [179].

Well-circumscribed inflammatory vesicular lesions have been described on the palm of a 2-year-old patient; the child’s brother had tinea capitis, with a history of animal contact [180]. This leads us to consider all clinical findings present in the patient’s environment.

Tinea pedis can mimic dyshidrotic eczema, shoe dermatitis, or juvenile plantar dermatitis [48,49].

Inflammatory tinea manuum and tinea pedis were reported to mimic bacterial cellulitis in adults and pediatric patients [50,180].

#### 5.3.4. Atypical Tinea Corporis

Tinea corporis is characterized by an annular plaque with advancing, raised, erythematous, scaling borders surrounding a clear center [5,78]. Extensive tinea corporis occurs mainly in patients with underlying immune disorders such as HIV or following systematic and topical use of steroids [148,181]. Immunosuppressed patients, including transplant recipients, cancer patients, and those on immunosuppressive therapy, are susceptible to deeper dermatophyte infections beyond the superficial layers [182].

Tinea corporis is a common and often overlooked infection that can also co-occur with other skin conditions [109]. Occasionally, tinea corporis simulates various conditions, including psoriasis, pityriasis rosea, secondary annular syphilis, nummular eczema, lupus erythematosus, and granuloma annulare [83,155,183]. The morphology, the facial distribution, and the recurrence of the fungal lesions at regular intervals after sun exposure are misleading signs of cutaneous lupus erythematous [83].

Additionally, tinea corporis is able to simulate leukocytoclastic vasculitis (LCV) [183], which is an inflammatory small vessel disease provoked by circulating immune complexes, following drug exposure or acute infections. Moreover, the histopathological characteristics of early-onset LCV are also visible in other conditions such as scurvy and tinea corporis [183].

Consequently, immunosuppression should be investigated and excluded in any patient with an atypical tinea corporis presentation [70].

#### 5.3.5. Tinea Recidivans

It represents another uncommon clinical entity that describes the presence of lesions at the periphery of a healed skin wound [123].

## 6. Atypical Type of Lesions

### 6.1. Lesions Configuration

#### 6.1.1. Annular Lesions

Unusual, large annular plaques can be found on the scalp, body, and flexures. The responsible species are usually, *Trichophyton rubum*, *Trichophyton mentagrophytes*, and *Microsporum gypseum* [107].

Typically, tinea presents as circular patches. However, in patients with ichthyosis, the lesions often appear less distinct and well-defined [63].

Tinea imbricata, called “tokelau” and “chimberê” in Brazil is a unique chronic variant of tinea corporis, caused by *T. concentricum* and characterized by impressive itchy squamous concentric circles, affecting large parts of the body [137]. Unexpectedly, *Trichophyton tonsurans* infection manifested as multiple concentric annular erythema in a Japanese documented case report [29]. Tinea incognito can mimic tinea imbricata, especially after prolonged use of corticosteroids (Figure 10).

Nonetheless, cases of concentric rings evocative of tinea imbricata in secondary syphilis exist in the literature [184].

#### 6.1.2. Nummular Lesions

Atypical cases of *Microsporum canis* were clinically described as nummular eczema, granuloma annulare, and granuloma faciale [124].

*Trichophyton verrucosum* has been reported to cause lesions clinically resembling nummular dermatitis in a Korean case report [185].

### 6.2. Lesions Color

#### 6.2.1. Erythematous Lesions

Erythematous lesions are non-specific for any particular dermatological condition and frequently accompany atypical superficial mycoses. This lack of specificity can pose diagnostic challenges. For instance, a 64-year-old woman with chronic myelocytic leukemia who developed erythematous hand lesions resembling eczema during chemotherapy for blastic crisis was ultimately diagnosed with *Trichosporon asahii* infection based on mycological results [136].

Different types of fungal erythemas can occur on various skin locations. Examples include the following:-Erythematous skin macules on arms and groin areas: this can be caused by Trichophyton species.-Erythema annulare on the trunk and upper limbs: this can be caused by *Trichophyton verrucosum* in adult males [121].-Multiple erythema annulare with slightly scaly plaques on the torso: this presentation can be associated with HIV, as seen in a woman diagnosed with the condition [78].-Erythema multiforme: this can be attributed to *Microsporum canis* dermatophytosis, as described by Alteras [124].

#### 6.2.2. Dyschromic Lesions

Skin hypochromic lesions can occur due to normal aging, environmental factors (cumulative sun exposure and microtrauma), inherited diseases (Ash leaf spots in tuberous sclerosis), nutritional deficiencies such as Kwashiorkor (protein malnutrition condition), and iron or vitamin B12 deficiencies.

Inflammatory causes such as vascular diseases, Bier’s spots, Pityriasis alba, post-viral exanthema, skin procedures (cryotherapy and dermabrasion), cosmetic inflammation, skin bleaching agents (hydroquinone), and burns.

Possible differential diagnosis of tinea versicolor will include pigmentary disorders such as vitiligo, idiopathic guttate hypomelanosis, and melasma; scaling is usually absent in these disorders. Other diseases to exclude are pityriasis alba, Hansen’s disease, pityriasis rosea, pityriasis rotunda, hypo- or hyperpigmented mycosis fungoides, secondary syphilis, lentigo solaris, piebaldism, and post-inflammatory hyperpigmentation [186].

Pityriasis alba often presents with small, hypopigmented patches on the face. The excessive-use of corticosteroid when treating eczema may induce this skin disease. In addition to other factors, sun and wind exposure and nutritional deficiencies (copper and iron) are also frequently incriminated.

The presence of greenish–yellow fluorescence under Wood’s light would strongly suggest tinea versicolor, while a negative result, associated to the absence of yeast on tape stripping examination, solidifies the diagnosis of pityriasis alba [187,188] (Figure 11).

Pityriasis rotunda is a rare disorder of keratinization characterized by persistent, hyperpigmented or hypopigmented, geometrically perfect circular patches of dry ichthyosiform scaling with no inflammatory changes. Pityriasis rotunda lesions may be associated with malignancies and liver diseases and it can be misdiagnosed as pityrirasis versicolor. In addition, some of the diseases also caused by *Malassezia* such as seborrheic dermatitis and confluent and reticulated papillomatosis of Gourgerot and Carteaud may coexist and or resemble pityrirasis versicolor, thus making diagnosis more difficult [189].

Although less typical, *Malassezia* can also target the face, scalp, and genital areas. Facial and penile lesions are relatively frequent in infants and immunocompromised individuals [190].

#### 6.2.3. Black Lesions

Tinea nigra commonly affects the palms of hands and the soles of feet [133].

Tinea nigra is an infrequent superficial fungal infection caused by a mold that exclusively affects the stratum corneum. It manifests as dark patches on the palms and soles, typically with minimal scaling [133]. The lesions have a noticeably darker border. Due to its dark coloration, tinea nigra can sometimes be mistaken for a melanocytic lesion. Dermoscopy can be used to distinguish between these two entities, as tinea nigra exhibits brown strands or “pigmented spicules” [191,192]. This fungal infection is caused by various molds, including Phaeoannellomyces, Hortaea, or *Exophiala werneckii* [133]. In Italy, a 13-year-old girl presented with a pigmented lesion on her left palm that appeared 2 years before, after a trip to a Greek island, the tinea nigra lesion was misinterpreted as melanocytic by dermatologists [193].

Black superficial lesions include melanocytic lesions such as melanoma-like lesions and black eschars caused by Mucorales. Moreover, a documented atypical case report of black dot ringworm lesions was due to *Trichophyton tonsurans* [120].

### 6.3. Lesions Morphology

#### 6.3.1. Urticaria

A possible coincidental association between urticaria and dermatophytosis was identified via the presence of *Epidemophyton floccosum* in skin scarping. However, urticarial lesions of undefined origin should be considered for a clinical picture of Id reaction and patients should be carefully investigated for superficial fungal infections [194].

#### 6.3.2. Purpuric Lesions

Reports in the literature also include rare forms of purpuric tinea corporis, presenting as erythematous, purpuric, scaling macules on a calf, caused by *Trichophyton violeceum* [114,185,195].

#### 6.3.3. Papular Lesions

Papular presentation is current in dermatophytosis. Due to the atypical presentations of superficial fungal infections in keratinization disorders, which can closely resemble other skin conditions, achieving a correct diagnosis can be complex and require careful consideration [12].

For example, both tinea corporis and congenital ichthyoses are common skin diseases. The association between these two conditions is plausible due to the immune and barrier defects present in ichthyoses [12].

On the other hand, fungal manifestations are defined as atypical when the inflammatory component is more severe, presenting follicular papules and pustules [17,183].

Fulminant papulopustular tinea corporis caused by *Trichophyton mentagrophytes* was described [85].

Interestingly, pityriasis rosea is a common viral exanthem, described as a limited papulosquamous eruption mimicking the common papular tinea corporis [196].

#### 6.3.4. Plaques

An 89-year-old woman presented with plaque-like lesions accompanied by pustules and desquamation on the back and front of the trunk for approximately one year, until the mycological examination revealed a tinea incognito [197].

#### 6.3.5. Vesicular/Bullous Lesions

Fungal infections can manifest as vesicular or even bullous tinea, where erythematous dermatitis presents with annular lesions and raised, blister-like borders [198,199].

Inflammation further promotes fungal colonization and can sometimes lead to vesicle formation at the edges of the affected area [34,35]. Additionally, the intense inflammation caused by the zoophilic fungus *Trichophyton mentagrophytes*, combined with a significant delayed-type hypersensitivity reaction, are believed to contribute to the development of vesicles or bullae in some cases.

Furthermore, secondary blistering may occur in cases of severe inflammation. The intensity of inflammation is influenced by the type of fungus, the patient’s immune system, and the degree of follicular involvement [200].

The atypical clinical cases are generally indicated by the large size of the lesion, and the presence of pustules in the central region [8]. Atypical very inflammatory bullous lesions due to *Trichophyton mentagrophytes* have been described in HIV individuals [75]. In addition, pustular manifestations are seen at the border of inflammatory erythematous lesions of tinea [123].

A 30-year-old woman with no underlying disease presented with an inflammatory, pustular, and crusted plaque on the pubis and vulva after shaving, which led to a diagnosis of tinea incognito caused by *Trichophyton mentagrophytes* [201].

Interestingly, the pustular forms of tinea incognito can mimic the appearance of pustular psoriasis especially among immunosuppressed individuals, so it is important to consider it as a possible diagnosis [107].

#### 6.3.6. Abscesses

Reported coexisting diseases that are suspected of inducing an immunosuppression and favoring dermatophytes abscesses are malignancy, renal and liver transplantation, CARD9 deficiency, collagen disease, nephrotic syndrome, bullous pemphigus, and diabetes mellitus. Most reports describe unsuspected, pre-existing superficial mycosis (Inaoki) [202]. Some of the abscessed dermatophytosis cases are cited in Table 4.

Multiples abscess of the lower extremities were provoked by *Trichophyton rubrum* [182].

#### 6.3.7. Nodular Lesions

Rare fungal nodular lesions are attributed to *Sporothrix* sp. In addition, fish tank granuloma is a rare skin infection caused by *Mycobacterium marinum*. It occurs after exposure of skin abrasions to contaminated water or infected fish. The most common presentation is a solitary nodule, often with sporotrichoid spread [6].

##### Majocchi’s Granuloma

In rare cases, topical steroid overuse can weaken local immunity, potentially allowing dermatophytes to invade the dermis. This can lead to pruritic papules, pustules, and nodules forming in areas with existing dermatophytic infections [123]. This invasive clinical picture is called “Majocchi granuloma” (Figure 12).

Immunocompromised patients (diabetes mellitus, chronic liver and kidney disorders, transplant recipients) are also more likely to develop deep dermatophytosis, a rare disorder characterized by invasion of dermatophytes into the dermis and subcutaneous tissue, rather than just keratinized skin [71].

Granulomatous lesions due to *Trichophyton mentagrophytes* and *Trichophyton rubrum* have mainly been described [209].

Moreover, an atypical case of *Microsporum canis* was described as granuloma annulare and granuloma faciale [124].

Bizarrely, an atypical presentation of Majocchi’s granuloma with multiple non-tender erythematous folliculo-centric nodules and central pustulation on the lower extremities has been documented. There was no history of trauma or application of topical corticosteroids over the affected area, and the patient denied shaving the affected area [210]. Some of the reported nodular forms of dermatophytosis are depicted in Table 5.

#### 6.3.8. Ulcerative Lesions

Cutaneous mucormycosis may be a primary disease following skin barrier breakage or may occur as a consequence of hematogenous dissemination from other sites, and the outcome of the disease is strictly dependent on the patients’ conditions. Primary cutaneous mucormycosis can involve the subcutaneous tissue as well as the fat, muscle, and fascial layers [216].

Patient’s skin necrosis resembling *Pyoderma gangrenosum*, especially if they are chronically immunocompromised, diabetic, severely malnourished, or treated with broad-spectrum antibiotics, should suggest the possibility of mucormycosis [96].

A unilateral ulcerative lesion on the hand resembling leishmaniasis but caused by Nannezia gypsea was identified in a 23-year-old female patient in Mexico [217].

#### 6.3.9. Tumoral Lesions

Fungal infections can coexist with or mimic malignancies, regardless of the site of the lesion, namely, the skin [1], nails, and scalp [34,102,156,165].

A pediatric patient with *Trichophyton verrucosum* infection presented with lesions clinically resembling a vascular tumor, likely a consequence of the exceptionally severe and diverse clinical presentations of zoophilic species [218].

There are a few documented cases of dermatophytosis mimicking Kaposi’s sarcoma reported in a HIV-positive patient [76,77,219]. A study highlighted that one month after stopping HIV medications, erythematous lesions steadily increased and spread to the legs, feet, abdomen, and buttocks of the patient, marking an extensive form of dermatophytosis [77].

Notably, melanized (dematiaceous) fungal infections due to *Nigrograna mackinnonia* can also simulate skin cancer in organ transplant recipients [220].

## 7. Conclusions

The rise in atypical superficial mycoses is a complex puzzle with multiple contributing factors.

To avoid diagnostic pitfalls, it is crucial to re-evaluate patients with chronic inflammatory dermatosis who do not improve with standard treatment or exhibit unusual clinical presentations, such as severe inflammation or extensive lesions.

A thorough history and physical examination can help rule out potential fungal infections.

## Figures and Tables

**Figure 1 jof-10-00295-f001:**
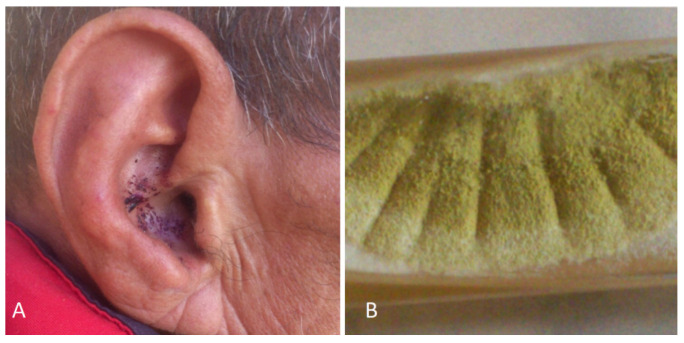
(**A**) Unexpected otomycosis after self-induced trauma in schizophrenic patient. (**B**) *Aspergillus flavus* macroscopy after ear swab culture.

**Figure 2 jof-10-00295-f002:**
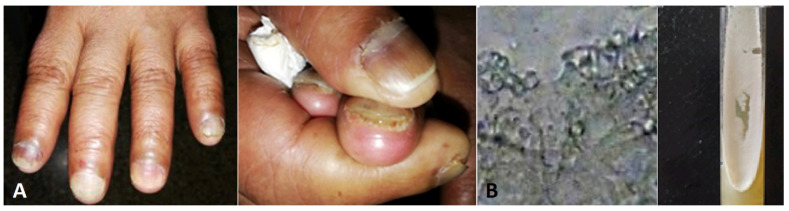
(**A**) Onycholyse associated with exudate and oedema in a patient under Docetaxel chemotherapy. (**B**) Budding yeast and creamy white colonies of *Candida guillermondii* on SDA.

**Figure 3 jof-10-00295-f003:**
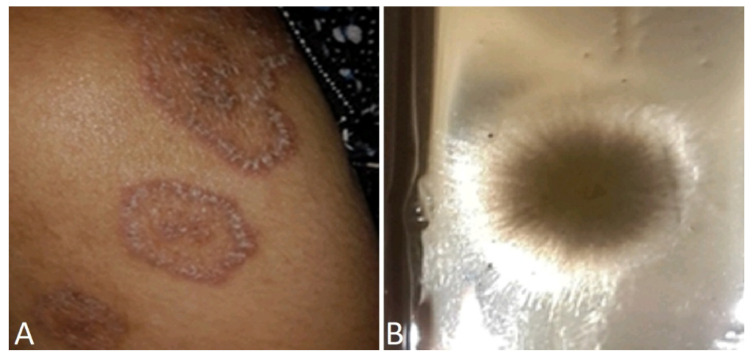
(**A**) *Microsporum canis* tinea corporis in female with HIV at day 7 after hospitalization for cryptococcus neuromeningitis. (**B**) macroscopic aspect of skin scraping culture.

**Figure 4 jof-10-00295-f004:**
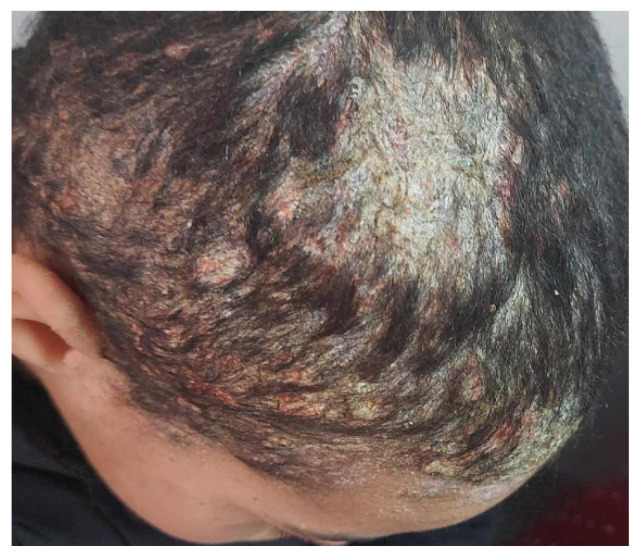
Tinea capitis due to *Microsorum canis* in a patient with underlying psoriasis vulgaris.

**Figure 5 jof-10-00295-f005:**
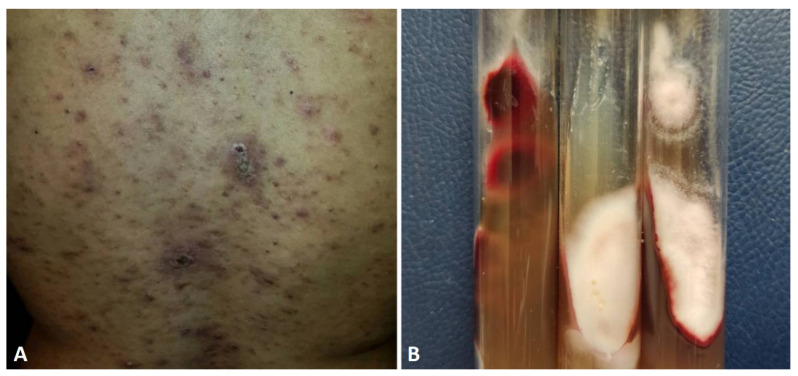
(**A**) Acne-like lesions on the back of a patient with Behcet’s disease. (**B**) *Trichophyton rubrum* culture after skin scraping.

**Figure 6 jof-10-00295-f006:**
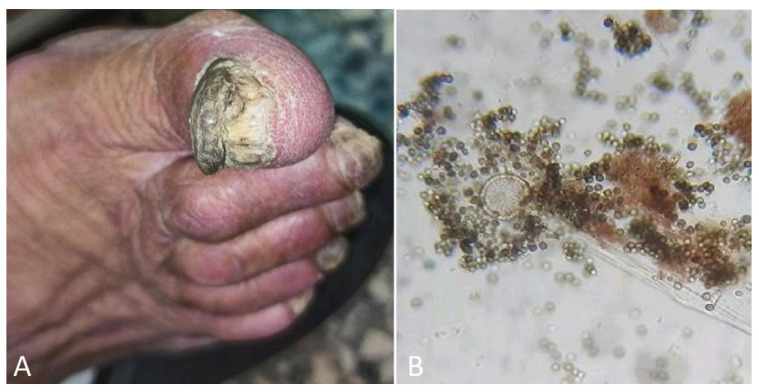
(**A**) Onychogryphosis in a 75-year-old patient. (**B**) *Aspergillus niger* microscopy after culture.

**Figure 7 jof-10-00295-f007:**
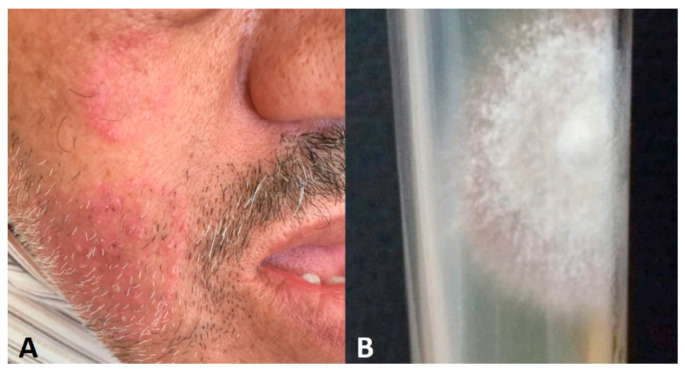
(**A**) Unexpected tinea faciei in acromegaly patient under corticosteroids. (**B**) *Trichophyton mentagrophytes* culture after skin scraping.

**Figure 8 jof-10-00295-f008:**
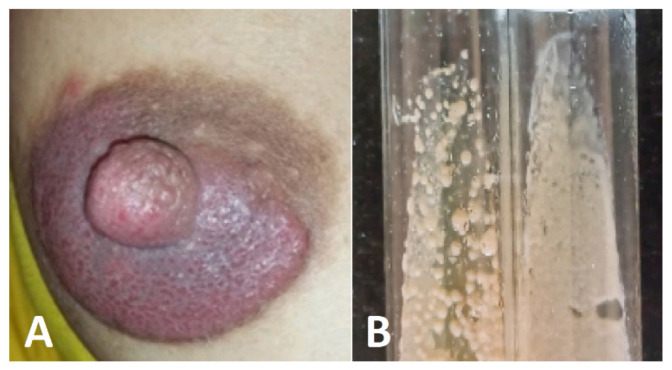
(**A**) *Candida albicans* breast hyperkeratosis and discharge. (**B**) Creamy white colonies of *Candida albicans*.

**Figure 9 jof-10-00295-f009:**
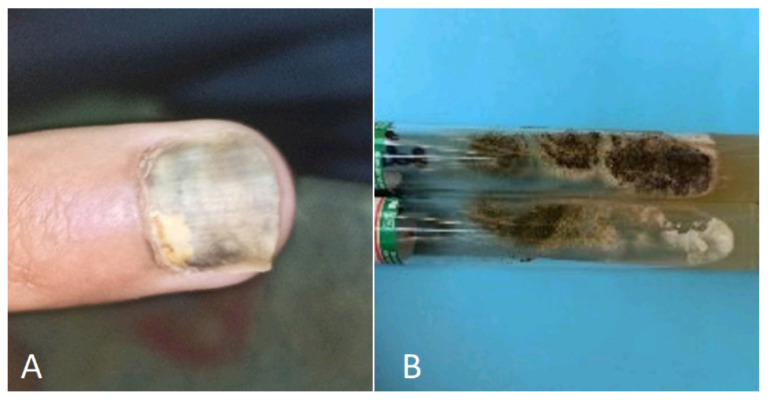
(**A**) black ungual discoloration in vegetable vendor, (**B**) *Aspergillus niger* culture after ungual debris culture.

**Figure 10 jof-10-00295-f010:**
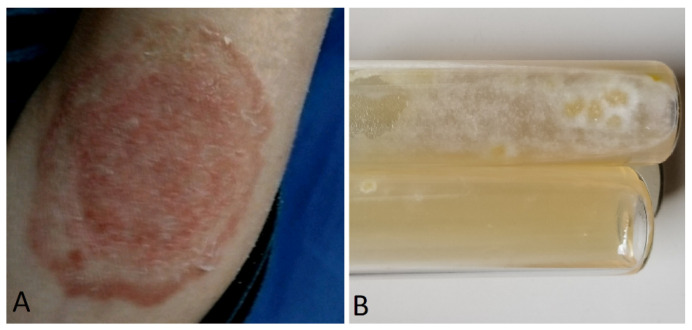
(**A**) *Microsporum canis* tina incognito mimicking tinea imbricata. (**B**) *Microsporum canis* culture result.

**Figure 11 jof-10-00295-f011:**
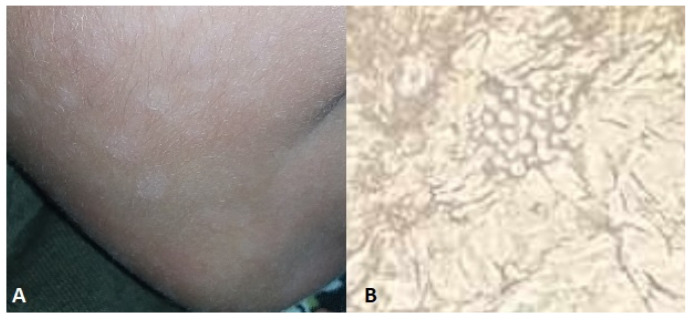
(**A**) Tinea versicolor misdiagnosed as *Pityriasis alba*. (**B**) microscopic aspect of *Malassezia* sp.

**Figure 12 jof-10-00295-f012:**
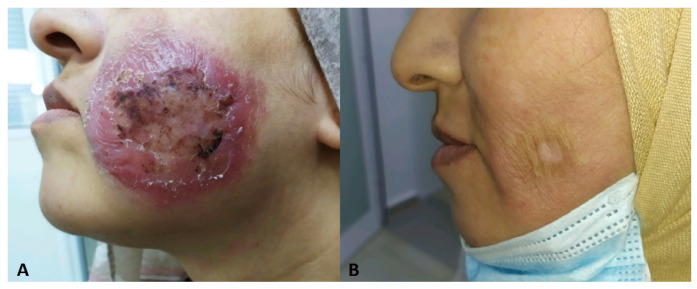
(**A**) Granular atypical dermatophytosis lesion before treatment, (**B**) the healing lesion after Terbinafine treatment.

**Table 1 jof-10-00295-t001:** Examples of atypical clinical presentations.

Description	S/Age	Localization	Fungi	Country	Reference
psoriasis-like	-	Body Palms Flexures	*T. rubrum* *T. mentagrophytes* *M. gypseum*	India	[107]
		Trunk, groin, buttocks	*T. rubrum*	India	[68]
		Lower extremities	*T. rubrum*	USA	[108]
	M/68	-	*-*	Slovenia	[109]
		Legs	*M. canis*	Serbia	[62]
		Body All Nails	*T. rubrum*	Switzerland	[10]
		Trunk, lower extremities, toenails	*T. rubrum*	New Zealand	[110]
Inverse psoriasis-like		Intertriginous areas	*Candida* sp.	India	[111]
Eczema-like		-	*T. mentagrophytes* *T. rubrum*	Korea	[90]
		Face	*T. rubrum*	India	[112]
	M/36	Arms Inguinal region	*Trichophyton eboreum*	Switzerland	[113]
	F/64	Hand	*Trichosporon asahii*	Japan	[114,115]
Seborrheic dermatitis-like	-	Face Scalp	*T. rubrum* *T. mentagrophytes* *T. tonsurans* *M. gypseum*	India	[107]
	M/77	Scalp Nails	*T. rubrum*	China	[116]
		Face Neck		India	[5]
Erythema-like		Torso	*-*	Grenada	[78]
lupus erythematosus-like	-	Face			[83,117]
Furoncle-like	-	Body	*T. tonsurans*		[107]
Acne-like	M/29	Back	*T. rubrum*	Algeria	[118]
Prurigo-like	-		*T. rubrum*	India	[107]
Ichthyosis-like		Body	*T. menta*	India	[107]
Contact dermatitis-like	M/12	Right eyebrow	*T. menta*	Chile	[119]
Impetigo-like	M/41	Right forearm	*T. tonsurans*	Japan	[120]
Allergic-like dermatitis	M/31	Trunk, upper limbs	*T. verrucosum*		[121]
Cellulite-like	4 cases		*-*	USA	[50]
Pityriasis rubra pilaris	M/54	Trunk, shoulders, upper arms	*Malassezia* sp.	USA	[122]
Mycosis fungoides-like	F/62	Right chin, Right elbow	*-*	USA	[1]

**Table 2 jof-10-00295-t002:** Examples of atypical location of superficial mycosis.

Infection Location	Presentation/Context	Agent	Reference
palms and fingernails	Achromic lesions	*Malassezia* sp.	[84]
Penis		*Malassezia* sp.	[138]
Axilla	Eryth and papular rash	*Malassezia* sp.	[139]
Eyebrows	Tinea of the scalps and eyebrows in diabetic elderly	*T. violaceum*	[140]
Right eyebrow	Erythematous lesion	*T. mentagrophytes*	[119]
Scalp and eyebrow	Tinea of the scalp and left eyebrow	*M. canis*	[141]
Eyelid and skin	Upper eyelid swelling for 6 monthsHIV patient with disseminated lesions	*Rhinosporidium* sp.	[142]
Lip	Dermatophyte presenting as a verrucous nodule of the lip	*T. rubrum*	[143]
Penis	Infection of the penis/after wearing penile clothe	*M. canis*	[103]

**Table 3 jof-10-00295-t003:** Examples of atypical onychomycoses.

Clinical Presentation	Clinical Details	Causative Agent	Reference
Brown pigmentation	Usually, diffuse lesions	*Aspergillus niger* *Alternaria alternata* *Scytalidium dimidiatum*	[156,160]
Longitudinal melanonychia	Distal lesions	*Trichophyton rubrum* *Candida humicola* *Candida albicans* *Candida parapsilosis* *Scytalidium dimidiatum* *Alternaris* sp. *Exophiala* sp. *Aspergillus niger*	[102,156,165]
Distal and lateral subungual onychomycosis		*Alternaria alternate* *Scytalidium* sp.	[166]
Black pigmentation	Superficial coloration Periungual inflammation Proximal nail	*Aspergillus niger*	[157]
Onychodystrophy	Nail discoloration		[160]
Onycholysis	Greenish–black discolorations	*Candida parapsilosis*	[167]
Subungual onychomycosis	Mimicking a glomus tumor	*Aspergillus niger*	[162]
Onychodystrophy	All fingernails and toenails involved	*Cladosporium* sp.	[7]

**Table 4 jof-10-00295-t004:** Examples of atypical abscessed dermatophytosis.

Localization	Sex/Age	Culture	Underlying Disease	Country	References
Lip	F/40	*T. rubrum*	-	India	[143]
Hand	F/52	*T. rubrum*	Polychondritis	Japan	[203]
Extremities, trunk, face	F/44	*T. rubrum*	Myasthenia gravis	Japan	[204]
Leg	M/54	*T. rubrum*	None	Japan	[202]
Thigh	F/24	*T. mentagrophytes*	None	USA	[95]
Scalp	F/19	*M. canis*	None		[205]
Upper extremities	M/52	*T. rubrum*	Immunosuppressive therapy	France	[206]
Malleolus	F/68	*T. rubrum*	-	Korea	[207]
Foot	M/39	*Trichophyton* sp.	Type-2 diabetes	USA	[208]

**Table 5 jof-10-00295-t005:** Examples of atypical nodular dermatophytosis.

Localization	Sex/Age	Culture	Underlying Disease	Country	References
Leg, trunk	M/28	*T. verrucosum*	Nephrotic syndrome	Japan	[209]
Groin	M/58	*T. rubrum*	Nephrotic syndrome	Japan	[211]
Extremities, truck, face	M/45	*T. rubrum*	Haemochromatosis	USA	[212]
Forearm	F/62	*T. tonsurans*	-	China	[213]
Breast	F/28	*T. violaceum*		India	[169]
Genitoinguinal	M/54	*-*	-	Danemark	[214]
Forearm	M/53	*T. rubrum*		Spain	[136]
Buttock	M/73	*T. rubrum*	Suppressive drugs	Japan	[215]

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
