# Peer review of "Atypical and Unpredictable Superficial Mycosis Presentations: A Narrative Review"

_jof, 2024, doi:10.3390/jof10040295_

Round 1

Reviewer 1 Report

Comments and Suggestions for Authors

Dear authors, thank you for your interesting paper. I humble suggest some minir issues:

- first sentence of introduction: not only at the first stage. For example, tinea incognito can be midsiagnosed also in a late stage. Also, diffuse tinea corporis can be misdiagnosed as psoriasis in a late stage.

- A paragraph about "Methods" should be added: how this narrative review has been performed?

- page 2. I think that "enviromental factors" should be a paragraph with a number (or not?) In this paragraph you could add tinea imbricata, or tokelau, diffuse in Oceania (Veraldi S, et al. Tinea Imbricata in an Italian Child and Review of the Literature. Mycopathologia. 2015)

- Did you find in literature cases of tinea with progressive course due to a wrong treatment or a concomitant with biological?

- About tinea nigra, a more recent dermoscopic description can be added: Nazzaro G et al. Tinea nigra: A diagnostic pitfall. J Am Acad Dermatol. 2016

Author Response

Thank you sir, the responses are attached in a word file

Reviewer 2 Report

Comments and Suggestions for Authors

This is an interesting paper. It is a very useful resource for clinicians. However, the article could be improved. 

Line 25 - change the word appearances to something like ways. Already have "appear" in the sentence

Line 30 - because of a lack of suspicion - best way to say it

Line 57 - there is a typographical error. "Some" not "some" at the start of the sentence. Also should say --- Some fungi are spread globally 

Line 101-102 - do you mean that Histoplasma and Coccidioides spp. require immunosuppression for dissemination? Need to clarify this sentence.

Line 110 - de should be the 

Line 138 - Furthermore is spelt incorrectly. Please correct.  

Line 141 - Aspergillus needs to be in italics. Please make sure that this fungus and all others are italicised throughout

Line 149 - Injury needs to be all in small letter - injury

Line 174 - dermatologists can assume is better. Gives room for this who do recoginse these things as unusual.  Would add the word can.

Line 176 - should be diagnostic confusion

Line 177 - should be rheumatoid

Line 199 - not sure how gut dysbiosis leads to immunosuppression. I would leave out the immunosuppression unless you can provide a strong link between gut dysbiosis and immunosuppression. Also how severe is the immunosuppression. I think unlikely enough to lead to serious fungal infection.   

Line 206 - what is LE? Probably best to spell it out in full.

Line 211 -indigenous drug. What does this mean? Maybe you didn't complete a paragraph? Please add a paragraph about this or delete the statement altogether

Line 220 - developed is spelt in correctly. Also do you mean the cutaneous manifestations of mucorales?   

Line 221 - deferoxamine - do you mean cutaneous manifestations. If this is a paper on superficial mycosis, you need to be specific.

Line 238 - typographical errors - should be cutaneous aspergillosis

Line 240 - how are ventilators implicated in cutaneous fungal infections? Need to say

Line 251 - needs to be Organ

Line 418 -  needs to be No evident

Line 425 - I think (Brown) should be deleted

Line 426 - disseminated is spelt incorrectly. Please correct.

Line 444 - should be  ---- An elderly woman....

Line 448 - should be----- Fungal rashes

Line 492 - chronic needs to be a small letter at the start 

Line 505 - nails needs to have a small letter at the start

Line 513 - Candida needs to be in italics

Paragraph on piedra - it would be good to add a photo on this as a visual example.   

Line 623 - should be simulate leukocytoclastic vasculitis 

Line 632 - do you mean a healed skin wound? Please make it clear. 

Line 641 -should have a small i -----imbricata

Line 698 - Malassezia, should be in italics

For the paragraph on dyschromic lesions - some images would be good. Even as Figures xa, xb, xc etc. 

Line 730 - should be example

Majocchi's granuloma - need to make it clear that it is due to dermatophytes. Not really clear from the paragraph on it as it stands

Comments on the Quality of English Language

Needs editing from a grammatical point of view. 

Author Response

Thank you sir for reviewing my article, i m honored, the responses are in the attached word file
